# T Cells Directed against the Metastatic Driver Chondromodulin-1 in Ewing Sarcoma: Comparative Engineering with CRISPR/Cas9 vs. Retroviral Gene Transfer for Adoptive Transfer

**DOI:** 10.3390/cancers14225485

**Published:** 2022-11-08

**Authors:** Busheng Xue, Kristina von Heyking, Hendrik Gassmann, Mansour Poorebrahim, Melanie Thiede, Kilian Schober, Josef Mautner, Julia Hauer, Jürgen Ruland, Dirk H. Busch, Uwe Thiel, Stefan E. G. Burdach

**Affiliations:** 1Department of Pediatrics, Children’s Cancer Research Center, Kinderklinik München Schwabing, School of Medicine, Technical University of Munich, 80804 Munich, Germany; 2Institute for Medical Microbiology, Immunology and Hygiene, School of Medicine, Technical University of Munich, 81674 Munich, Germany; 3Department of Gene Vectors, Helmholtz Centre Munich, 81377 Munich, Germany; 4DZIF, German Center for Infection Research, Partner Site Munich, Germany Institute of Clinical, 81675 Munich, Germany; 5Munich Childhood Health Alliance (CHANCE) e.V, 80337 Munich, Germany; 6DKTK German Cancer Consortium, Partner Site Munich, 81675 Munich, Germany; 7Institute of Chemistry and Pathobiochemistry, TUM School of Medicine, Technical University of Munich, 81675 Munich, Germany; 8Center for Translational Cancer Research (TranslaTUM), 81675 Munich, Germany; 9Translational Pediatric Cancer Research-Institute of Pathology, School of Medicine, Technical University of Munich, 81675 Munich, Germany; 10Department of Molecular Oncology, British Columbia Cancer Research Centre and Academy of Translational Medicine, University of British Columbia, Vancouver, BC V5Z 1L3, Canada

**Keywords:** Ewing sarcoma, chondromodulin-1, immunotherapy, orthotopic TCR replacement, CRISPR/Cas9, retroviral transduction

## Abstract

**Simple Summary:**

The canonical methods of TCR gene delivery in pre-clinical and clinical applications are based on viral transduction of full-coding sequences, including α- and β-chains recognizing tumor-specific antigens and tumor-associated antigens. As the transduced α- and β-chains may mispair with the endogenous α- and β-chains, the resultant new antigen specificities may cause auto-reactivity, potentially leading to graft-versus-host disease. The mispaired TCRs may also lose their function. We assessed the feasibility of endogenous TCR orthotopic replacement with a TCR containing a CHM1 targeting sequence via CRISPR/Cas9, evaluated tumor recognition and cytotoxicity function of the CRISPR/Cas9-engineered T cells; compared the prevention of endogenous TCR expression in CRISPR/Cas9 vs. retrovirally engineered T cells. We show that both engineered T cell products specifically recognize tumor cells and elicit cytotoxicity in vitro, with CRISPR/Cas9 engineered T cells providing a more prolonged cytotoxic activity.

**Abstract:**

Ewing sarcoma (EwS) is a highly malignant sarcoma of bone and soft tissue with early metastatic spread and an age peak in early puberty. The prognosis in advanced stages is still dismal, and the long-term effects of established therapies are severe. Efficacious targeted therapies are urgently needed. Our previous work has provided preliminary safety and efficacy data utilizing T cell receptor (TCR) transgenic T cells, generated by retroviral gene transfer, targeting HLA-restricted peptides on the tumor cell derived from metastatic drivers. Here, we compared T cells engineered with either CRISPR/Cas9 or retroviral gene transfer. Firstly, we confirmed the feasibility of the orthotopic replacement of the endogenous TCR by CRISPR/Cas9 with a TCR targeting our canonical metastatic driver chondromodulin-1 (CHM1). CRISPR/Cas9-engineered T cell products specifically recognized and killed HLA-A*02:01+ EwS cell lines. The efficiency of retroviral transduction was higher compared to CRISPR/Cas9 gene editing. Both engineered T cell products specifically recognized tumor cells and elicited cytotoxicity, with CRISPR/Cas9 engineered T cells providing prolonged cytotoxic activity. In conclusion, T cells engineered with CRISPR/Cas9 could be feasible for immunotherapy of EwS and may have the advantage of more prolonged cytotoxic activity, as compared to T cells engineered with retroviral gene transfer.

## 1. Introduction

Ewing sarcoma (EwS) is a highly malignant bone and soft tissue cancer in children and adolescents characterized by early metastasis [1,2]. Primary EwS is treated by a combination of chemotherapy and surgery and/or radiation [3]. Patients with metastatic and refractory disease have been treated with extended radiation and high-dose chemotherapy, which is efficacious only in subgroups [4,5,6]. Moreover, current therapies are associated with acute and chronic adverse effects that compromise the quality of life in survivors [1] by chemotherapy-associated myeloid dysplastic syndrome, leukemia, and radiation-associated sarcoma. At the age of 50 years, 60% of childhood cancer survivors either died of long-term treatment effects or suffer from life threatening conditions [7]. No novel therapeutic modalities have been successfully introduced into the standard care of advanced EwS in the last 40 years. The overall survival, thus, remains unsatisfactory, especially in patients suffering metastasis or early relapse who have a 5-year overall survival < 30% [8]; novel therapeutic approaches are in urgent need.

The T cell receptor (TCR) can recognize peptide antigens presented on the cell membrane of the host cells by the histocompatibility complex (MHC)/human leukocyte antigen (HLA) system [9]. TCR is a heterodimer comprised most commonly of an α- and a β-chain [10], or alternatively of a γ and a δ chain [11]. TCR-based adoptive therapy allows the genetic redirection of the T cell specificity. Transduction with viral vectors is the conventional method of antigen-specific TCR insertion by either retro- or the lentivirus particles [12]. However, the random insertion of viruses into the genome raised safety concerns about insertional mutations and tumorigenesis, albeit the latter has been mainly observed in stem cells [13]. An unusual case has been described in T cells, where CAR-T cells originated from a single clone in which lentiviral vector-mediated insertion of the CAR transgene disrupted the TET2 gene and improved the expansion of the therapeutic clone to cause leukemia remission [14].

CRISPR/Cas9 engineered orthotopic TCR replacement leads to accurate α and β chain pairing, and regulation of the transgenic TCR is similar to that of physiological T cells [15]. The canonical methods of TCR gene delivery in pre-clinical and clinical applications are based on viral transduction of full-coding sequences, including α- and β-chains recognizing tumor-specific antigens (TSA) and tumor-associated antigens (TAA) [16,17]. As the transduced α- and β-chains may mispair with the endogenous α- and β-chains, the resultant new antigen specificities may cause auto-reactivity, potentially leading to graft-versus-host disease (GvHD). The mispaired TCRs may also lose their function. Albeit, we did not see relevant GvHD in our patients up to now, the number of patients is low [18], and the mechanisms are not fully understood. Schober et al. [15] at our university established a non-viral TRBC knock-out/TRAC knock-in model, which displayed a TCR regulation pattern very similar to that of a physiological T cell population [19,20,21].

The study reported here was initiated to compare T cells against the metastatic driver of EwS CHM1 engineered with CRISPR/Cas9 vs. retroviral gene transfer. In general, we asked whether CRISPR/Cas9 engineered T cell receptor insertion to the TRAC locus of CD3+ T cells preserves physiological properties and yields a therapeutic product that is at least as efficacious in immunotherapy of EwS as the product generated by retroviral gene transfer. Specifically, we assessed (1) the feasibility of an orthotopic replacement of the endogenous TCR with a TCR containing a CHM1 targeting sequence, (2) TCR expression, as well as tumor recognition and cytotoxicity function of CRISPR/Cas9-engineered T cells, (3) comparative prevention of endogenous TCR expression in CRISPR/Cas9 vs. retrovirally engineered T cells, and finally (4) characterization of CHM1 as a unique target. Figure 1 is a graphical abstract; the workflow of the study is depicted in Appendix A.

## 2. Materials and Methods

### 2.1. Cell Lines

The cell lines used in this work were described previously [22,23]. EwS cell lines and K562 were cultured in RPMI 1640 medium (Life Technologies Limited, Paisley, UK) with 10% fetal bovine serum (FBS) (Life Technologies Limited, Paisley, UK) and serum containing 100 U/mL penicillin and 100 ug/mL streptomycin (Life Technologies Corporation, Grand Island, NY, USA). For LCL and T2 cells, 1 mM Na-pyruvate and 1 mM non-essential amino acids (Life Technologies Limited, Paisley, UK) were also added into the medium. The 293Vec-RD114 packaging cells were cultured in DMEM (Life Technologies Limited, Paisley, UK) with 10% fetal bovine serum (FBS) (Life Technologies Limited, Paisley, UK) containing 100 U/mL penicillin and 100 ug/mL streptomycin (Life Technologies Corporation, Grand Island, NY, USA), and 1 mM Na-pyruvate and 1 mM non-essential amino acids (Life Technologies Limited, Paisley, UK). T cells were cultured in AIM-V (Life Technologies Limited, Darmstadt, Germany) medium with 5% human-AB serum (SIGMA-ALDRICH CHEMIE GmbH, Steinheim, Germany) containing 100 U/mL penicillin and 100 ug/mL streptomycin (Life Technologies Corporation, Grand Island, NY, USA). PBMCs were isolated by density-gradient centrifugation with Ficoll-Paque (GE Healthcare, Uppsala, Sweden), according to supplier’s instructions from healthy donor, as described previously [24]. Healthy donor blood samples were purchased from DRK-Blutspendedienst (Baden-Wuerttemberg-Hessen, Ulm, Germany; obtained after IRB approval and informed consent).

### 2.2. Expansion of TCR-Transgenic T Cells

After being purified with anti-PE magnetic beads (Miltenyi Biotec, Bergisch Gladbach, Germany) coupled to PE-mTCR antibody (Biolegend, San Diego, CA, USA), CHM1^319^-TCR-transgenic T cells were cultured in 25 cm^2^ cell culture flasks (TPP, Trasasingen, Switzerland). Cells were cultured with 25 mL T cell medium supplemented with OKT3 (50 ng/mL, Biolegend, San Diego, CA, USA), IL-2 (100 IU/mL, R&D Systems, Minneapolis, MI, USA), IL-7 (5 ng/mL, R&D Systems, Minneapolis, MI, USA), and IL-15 (2 ng/mL, R&D Systems, Minneapolis, MI, USA). Interleukins were added every other day. Irradiated LCL (100 Gy, 5 × 10^6^) and PBMCs (peripheral blood mononuclear cells) (30 Gy, 2.5 × 10^7^) pooled from at least three different donors are added as feeder cells.

### 2.3. Functional Characterization of CHM1^319^/HLA-A*02:01-Specific TCR Transgenic T Cells

T cell specificity was confirmed with interferon-γ (IFNγ)-Elispot assay (Mabtech AB, Nacka Strand, Sweden), according to the manufacturer’s information. A variable of 500 to 1 × 10^4^ T cells were used targeting 2 × 10^4^ EwS; K562 or T2 cells co-cultured for 20 h at 37 °C, 5% CO_2_, Elispot Reader (machine and software version 5.0; Advanced Imaging Devices GmbH, Straßberg, Germany) were used for detection.

T cell-mediated cytotoxicity was monitored with the impedance xCELLigence assay (Roche Diagnostics, Penzberg, Germany), allowing continuous measurement of T cell activity against target cell lines, including A673 and SK-N-MC. The 1 × 10^4^ A673 or 3 × 10^4^ SK-N-MC cells were plated 48 h before the addition of 5 × 10^3^ T cells.

### 2.4. Western Blot

After being washed once with PBS, the product was harvested and solubilized in RIPA lysis buffer (Abcam, ab156034, Waltham, MA USA) with protease inhibitor (Complete mini, Roche Diagnostic, Mannheim, Germany). Protein concentration was determined by BCA (Thermo Fisher Scientific, Ulm, Germany). Ten to fifty micrograms of protein extract were resolved on 10% SDS-PAGE and transferred onto PVDF membrane (Thermo Fisher Scientific, Dreieich, Germany). Primary antibody included PARP (Cell Signaling Technology, Danvers, MA, USA). GAPDH (Santa Cruz Biotechnology, Dallas, TX, USA) served as control. Second antibody used Anti-Rabbit (Santa Cruz Biotechnology, Dallas, TX, USA) and Anti-Mouse antibodies (Santa Cruz Biotechnology, Dallas, TX, USA). Detection was performed with ECL chemiluminescence reagent (Amersham Biosciences, Little Chalfont, UK).

### 2.5. TCR DNA Template Design

DNA template was synthesized by GeneArt (Life Technologies, Thermo Fisher Scientific, Dreieich, Germany). The DNA structure of CRISPR/Cas9-mediated HDR had 5′ homology arm (300–400 bp pairs), P2A, TCR-β, T2A, TCR-α, and bGHpA tail, as described previously [15]. The related sequences are available as Appendix A.

### 2.6. CRISPR/Cas9 Mediated TCR KI and Retrovirus Transduction

For fresh PBMC from buffy coat, CRISPR/Cas9-engineered endogenous TCR KO with or without exogenous TCR insertion was performed two days after T cell activation with CD3/28 Dynabeads^TM^ (Thermo Fisher Scientific Baltics UAB, Vilnius, Lithuania) and 100 IU/mL IL-2, 5 ng/mL IL-7, and 5 ng/mL IL-15 for 2 days. For frozen PBMC, thaw the cells and culture them with T cell medium plus 50 IU/mL IL-2 for one day before activation. The murinized and codon-optimized TCR construct (pMP71_CHM1_mu_opt) in retrovirus group was used, as it was previously [22]. The packaging cell line 293Vec-RD114^TM^ was seeded 24 h before transfection into six well plates. Transfection of plasmid for the production of retrovirus was performed using TransIT-293 (Mirus Bio LLC, Maison, WI, USA).

### 2.7. Analysis of Published Chip-Sequence Data and Microarray

ChIP-sequence data (GSE61944: GSM1517546, GSM1517547, GSM1517555, GSM1517556, GSM1517569, GSM1517570, GSM15175472, GSM1517573, GSM1517577, GSM1517581) were downloaded from the GEO database, and processed and displayed in the IGV browser [25]. Expression of CHM1 in EwS and bone marrow mesenchymal stem cell was mined from GEO database (GSE17618 and GSE6691), CCLE [26], and ProteomicsDB [27].

### 2.8. Statistical Analysis

GraphPad Prism (GraphPad Software, San Diego, CA, USA) was used to calculate mean and standard deviation of the mean (SD). Differences between groups were determined using the unpaired two-tailed Student’s *t*-test with *p*-values < 0.05 being considered statistically significant (* *p* < 0.05; ** *p* < 0.01; *** *p* < 0.001; **** *p* < 0.0001).

## 3. Results

### 3.1. Feasibility of Orthotopic Replacement of the Endogenous T Cell Receptor with a T Cell Receptor Containing Chondromodulin-1 Targeting Sequence

#### 3.1.1. CRISPR/Cas9-Engineered Orthotopic TCR Replacement

Based on our previous work [18] on immunotherapy of EwS, we focused on targeting the chondromodulin-1 peptide 319 (CHM1^319^) VIMPCSWWV. The T cell receptor (TCR) DNA template containing the sequence targeting the CHM1^319^ peptide [22] was established for homology-directed repair (HDR) (Appendix A). We performed PCR to amplify the knock-in (KI) fragment from the right homology arm to the left homology arm (Appendix A) to generate an abundant PCR product.

Next, we accomplished CRISPR/Cas9-engineered knock-out (KO) of the endogenous T cell receptor (hTCR), combined with or without CHM1^319^-TCR insertion into lymphocytes from peripheral blood mononuclear cells (PBMC). Single α- or β-strand, as well as double-strand KO, result in the loss of endogenous TCR surface expression (Figure 2A). Endogenous TCR KO combined with CHM1^319^-TCR insertion leads to a T cell population containing a murinized TCR (mTCR), which is hTCR negative, indicating successful CRISPR/Cas9-engineered gene editing. The KO efficacy was approximately 98.5%. In contrast, the KI efficacy in T cells from thawed T cells ranged between 11–23% (21% in Figure 1A, corresponding isotype staining is shown in Appendix A), while the efficiency of KI in fresh T cells reached 45% (Figure 2B, corresponding isotype staining is shown in Appendix A).

The CD3 complex is a heterodimeric glycoprotein cooperating with the TCR to convey signal transduction upon interactions with the antigenic peptides [28]. Upon combined TCR KO and KI, we see CD3 surface expression only in the population, where KI was successful, as indicated by Vβ23 (Vβ23-PE, Figure 2C, right panel, Q2) while the KI negative population remains CD3 negative (Figure 2C, right panel, Q4, corresponding isotype is shown in Appendix A). This indicates that CD3 expression is linked to TCR expression, which reversely indicates a successful CRISPR/Cas9-engineered gene editing.

#### 3.1.2. Tumor Recognition and Cytotoxicity by CRISPR/Cas9-Engineered T Cells

For functional analysis of CRISPR/Cas9-engineered T cells, we assessed T cell activation by IFNγ-Elispot and tumor cell apoptosis (cleaved-PARP) by Western blot. Selection of engineered T cells from six donors was initiated utilizing anti-murine TCR antibody coupled beads. The expression of the murinized TCR sequence (mTCR) was assessed by FACS analysis, yielding a 92.7% homogenous transgenic product in a representative experiment. (Figure 2D, right panel, Q2). CHM1^319^-restricted TCR transgenic T cells, specifically recognized T2 cells loaded with CHM1^319^ peptide, while T2 cells loaded with an HLA-A*02:0- binding influenza control peptide (FLU) were not recognized (Figure 2E, left panel). Cl-PARP as a parameter of apoptosis was specifically induced by T cells with orthotopic TCR replacement in HLA-A*02:01+ A673 but not in the HLA-A*02:01- SK-N-MC line (Figure 2E, right panel and Appendix A). Some marginal and variable cl-PARP was still seen after co-culture with TCR KO cells.

CRISPR/Cas9-engineered T cells secreted IFNγ when co-cultured with the HLA-A*02:01+ A673 and TC-71 EwS cell lines. In contrast, when co-cultured with HLA-A*02:01- cell lines SB-KMS-KS or SK-N-MC, only marginal IFNγ release was observed (Figure 2F). These findings indicate that CRISPR/Cas9-engineered T cells caused specific HLA-restricted EwS cell line recognition.

### 3.2. Higher Efficiency of Retroviral Transduction Compared to Gene Editing by CRISPR/Cas9

#### 3.2.1. TCR Transgenic T Cells Engineered by CRISPR/cas9 vs. Retroviral Gene Transduction

The experimental design is described in detail in the method section, as well as Appendix A. In brief, after isolating PBMC from buffy coat, we stimulated T cells with CD3/CD28 Dynabeads for two days. Meanwhile, we amplified the KI-DNA fragment for CRISPR/Cas9 transduction and transfected 293Vec-RD114 packaging cells with the pMP71-CHM1-TCR plasmid for retrovirus production. Subsequently, we purified the KI-DNA fragment for CRISPR/Cas9 or harvested the retrovirus for transduction. Next, we isolated the transgenic T cells with anti-mTCR antibody (mTCR-PE) and expanded the transgenic T cells for further functional analysis and in vivo experiments.

#### 3.2.2. Higher Efficiency of Retroviral Transduction Compared to Gene Editing by CRISPR/Cas9

We first assessed retroviral transduction efficacy. After transduction of T cells from two donors with GFP-containing control vector pMP71-GFP, we checked the GFP expression by fluorescence microscopy (Figure 3A) and by FACS analysis (Figure 3B). The cells from donor 1 were thawed, whereas cells from donor 2 were fresh. The transduction rate of thawed T cells was 78% (Figure 3B, upper panel), while fresh T cells reached 90% (Figure 3B, lower panel).

When comparing CRISPR/Cas9 orthotopic single gene replacement and multiple random insertions by retroviral transduction, we found retroviral transduction to be consistently higher. The efficacy of endogenous TCR orthotopic replacement with CHM1^319^-TCR ranged from 11% to 45%, whereas efficacy of retroviral transduction mainly ranged from 70% to 90%. Figure 3C,D depict the results of non-engineered CRISPR/Cas9 and retrovirus transduced T cells derived from the same donor. Retrovirus transduction efficacy was 77% (Figure 3C, lower panel Q2 plus Q3), whereas CRISPR/Cas9 transduction efficacy was only 19% (Figure 3C, middle panel Q2 plus Q3). As expected, the replacement of the endogenous TCR was more efficient in the CRISPR/Cas9 (Figure 3D, middle panel Q5 plus Q6), as compared to the retrovirus (Figure 3D, lower panel Q5 plus Q6) group. Corresponding isotype staining is shown in Appendix A. Figure 3E shows the transduction efficacy via CRISPR/Cas9 or retrovirus. Although there is biological variability, possibly donor-dependent, the efficacy is significantly higher for retrovirus, as compared CRISPR/Cas9 (*p* < 0.0001).

### 3.3. Prevention of Endogenous TCR Expression in CRISPR/Cas9 vs. Retrovirally Engineered T Cells

#### 3.3.1. Requirement of High Retroviral Gene Transduction Efficacy and High CRISPR/Cas9 KO Efficacy for Prevention of Endogenous TCR Expression and TCR Chain Mispairing

One of the advantages of orthotopic TCR replacement by CRISPR/Cas9 is that it avoids the mispairing of endogenous and exogenous TCR chains, and thus averts the generation of promiscuous TCRs recognizing off-target antigens. To gauge this postulated advantage of CRISPR/Cas9 vs. retroviral engineering, we compared the expression of endogenous TCR after CRISPR/Cas9 gene editing vs. retroviral transduction. We found that the decrease in endogenous TCR surface expression in the retrovirus group was similar to the expression in the CRISPR/Cas9 group (Figure 4A and Appendix A). After CHM1^319^-TCR transduction via retrovirus, the transgene positive population (red curve, Figure 4B) shows less endogenous TCR transgene, as compared to the negative population (blue curve, Figure 4B).

We also noted that repression of the endogenous receptor after retroviral transduction depends on transduction efficacy (Figure 4C). In this experiment, the retroviral transduction efficacy was only 42%. In the setting of this low transduction efficacy, we can identify two distinct subpopulations (Figure 4C, left panel): the upper cloud represents a subpopulation with high mTCR expression, i.e., the transduced subpopulation, whereas the lower cloud represents a population with a low mTCR expression. The high mTCR expressing subpopulation has a lower hTCR expression, as compared to the subpopulation represented by the lower cloud, which is characterized by low mTCR and higher hTCR expression. This low mTCR/high hTCR subpopulation is comprised of non-transduced cells, as indicated by low mTCR expression. Although there is some overlap between both subpopulations, the peaks of transduced and non-transduced subpopulations are distinct (Figure 4C, right panel). Isotype control is depicted in Appendix A. This finding implicates that a low retroviral transduction efficacy will yield a heterogenous product containing a large subpopulation at risk for mispairing and causing autoimmune side effects. Moreover, retrovirally transduced T cells express slightly more TCR, as documented by CD3 expression (Figure 4D and Appendix A), due to multiple gene copies, possibly because CD3 and TCR work as a protein complex for intracellular signaling transduction.

#### 3.3.2. Failure in KO of Endogenous β Chain Generates a Subpopulation with TCR Misparing

For clinical application, we have to ensure the KO of both endogenous TCR chains. Failure of β chain KO constitutes a risk of mispairing the transduced α with the endogenous β chain. The ratio of α to β chains in a single CRISPR/Cas9-engineered cell, where TRBC KO did not work, would be 2:1, since both transgenic chains are expressed from the α locus, in addition to the endogenous β chain expressed from the TRBC locus. When comparing the transduction efficacy of both procedures, in retrovirally transduced T cells the amount of transduced TCRs per cell are, by definition and by observation, higher (*n* > 1) than the single endogenous TCR (*n* = 1). On the other hand, if CRISPR/Cas9 KO is suboptimal, the risk of mispairing between the exogenous α with the endogenous β chains in a single cell, where TRBC KO did not work, would be X:1, with X being > 1. This implies that the risk of mispairing may be higher in CRISPR/Cas9, as compared to retrovirally transduced T cells, in case the KO is incomplete (Figure 4E). Failure in the knockout of the endogenous β chain leads to a subgroup of T cells expressing both the transgenic TCR and a mixed TCR with the endogenous β chain and the transgenic α chain. This is most likely the reason why we see some cells besides the main population in Figure 4E. We can also see this phenomenon in the right panel of Figure 2A. In Q2, we see a 0.76 population of the whole engineered T cells expressing both “mTCR“ and “hTCR“; these cells are also a result of the failure of the knockout of the endogenous β chain.

### 3.4. Specific Tumor Cell Recognition and Cytotoxicity In Vitro by Both T Cell Products with Better Prolonged Activity of CRISPR/Cas9-Engineered T Cells

We next assessed the cytotoxic effects on and specific recognition of EwS cell lines by transgenic T cells obtained with either orthotopic TCR replacement by CRISPR/Cas9 or random TCR placement by retroviral transfer. After isolation of the transgenic T cell with mTCR antibody, we co-cultured the T cells with HLA-A*02:01+ A673. Here, we identified an increase in the cl-PARP in A673 after co-culture with both transgenic T cell products. Furthermore, cl-PARP1 was observed with retrovirally transduced T cells, compared to CRISPR/Cas9-engineered T cells (Figure 5A and Appendix A). Of note, there was also an effect of the non-engineered T cells, probably due to non-specific allo-response. Non-engineered cells retain their endogenous HLA TCR, recognizing the tumor HLA-disparate haplotype.

To assess specific recognition, we performed an IFNγ-Elispot on day 35 and day 54 after T cell culture by co-culture T cells with T2 cells loaded with either CHM1^319^ peptide or with control peptide (FLU). On day 35, after T cell culture, we observed a higher IFNγ release with retrovirally transduced T cells, as compared to T cells with orthotopic TCR replacement (Figure 5B, left panel). In contrast, a higher IFNγ release was induced by T cells with orthotopic TCR replacement on day 54 (Figure 5B, right panel). This could be associated with or due to phenotypic differences (data not shown). Of note, we identified similar trends with the control peptide (Figure 5B), albeit with significantly weaker IFNγ signals.

We then performed xCELLigence assays to compare the cytotoxic effect on EwS cells with transgenic T cells on day 20. Both CRISPR/Cas9-engineered and retrovirus-transduced T cells lead to the significant killing, as measured by the detachment of the HLA- A*02:01+ A673 cell line without significant difference, but not to the killing of the HLA-A*02:01- SK-N-MC line, indicating HLA-A-restricted specificity (Figure 5C).

### 3.5. CHM1 as a Unique Immunotherapy Target in EwS

#### CHM1 Is a Direct Target of EWS-FLI1 Selectively Expressed in EwS

We have previously shown, EWS-FLI1 binds to the promotor and activates the transcription of CHM1 in EwS. Furthermore, CHM1 sustains the undifferentiated and invasive phenotype of EwS, which promotes lung metastasis of EwS [29]. It is highly expressed and required for metastasis [29] and serves as an EwS-specific antigen [18,22,30].

We first analyzed public CHIP-sequence data using the Integrative Genomics Viewer (IGV). CHIP-sequence data confirmed that EWS-FLI1 binds to two promotor sites of CHM1 and induces acetylation of H3K27 (H3K27ac) at both sites, which is associated with the activation of transcription (Appendix A). Forced expression of EWS-FLI1 in mesenchymal stem cell also enhances H3K27ac at the same sites (Appendix A).

We next mined the Cancer Cell Line Encyclopedia (CCLE) [31]. We found CHM1 mRNA is most highly expressed in EwS among all tumor cell lines (Appendix A). The Public Gene Expression Omnibus (GEO) database indicates that its expression in EwS tissues is significantly higher than in bone marrow mesenchymal stem cells (Appendix A). These results are in correspondence to our previous publications [29,32,33]. In addition, CHM1 expression does not correlate significantly with recurrence or metastasis (Appendix A).

Finally, we analyzed the expression of CHM1 at the protein level in normal tissue utilizing ProteomicsDB, developed by the Chair of Proteomics and Bioanalytics at the Technische Universität München and Cellzome GmbH [34,35]. We found only a very low expression in vitreous humor, lung, and heart and no expression in other tissues (Appendix A).

## 4. Discussion

### 4.1. TCR-Based Immunotherapy of Ewing Sarcoma

There is an obvious medical need for novel therapies in advanced Ewing sarcoma (AES), i.e., in patients with bone or bone marrow metastasis or early relapse. High dose therapies with autologous hematopoietic stem-cell rescues have only been beneficial in selected subgroups [4,36]. Allogeneic hematopoietic stem-cell transplantation (allo-SCT) from healthy donors, as efficacious immunotherapy of leukemia, has offered hints for beneficial effects in solid tumors [37], possibly also in AES patients [4,36,38]. However, no difference in survival with reduced- versus high-intensity conditioning before allo-SCT [39] could be detected. There is also no difference in survival after HLA-mismatched versus HLA-matched allo-SCT [38]. These findings imply that allo-SCT is not sufficient for immunotherapy of AES, and novel therapeutic strategies are in urgent demand, such as TCR-based immunotherapy [40]. CAR T cells have also been developed for AES [41]. The limitation of this approach is that CARTs can only target membrane molecules. However, molecules that are indispensable for tumor metastasis do not necessarily happen to sit on the cell membrane. Tumor metastasis, however, is what immunotherapy has to address, for two reasons: (1) Most local tumors are not a therapeutic challenge anymore with modern precision therapies—including theranostics, functional imaging, and in situ biomarker-guided surgery—or proton or particle therapy (2) Most cancer patients do not die of the tumor but of metastasis. With TCR-based immunotherapy targeting tumor-associated antigens and metastatic drivers of EwS, such as CHM1 [22,33], STEAP1 [24], PAPPA [42], and PRAME [43], our group previously achieved efficacious in vitro and in vivo cytotoxic targeting of HLA-A*02:01+ EwS. TCR-based immunotherapy even led to partial regression without GvHD in refractory HLA-A2+ patients [18]. TCR-based adoptive therapy also showed promising anti-sarcoma effects by targeting NY-ESO-1, leading to objective clinical responses [44]. More than 600 clinical trials about TCR-based immunotherapy are in processing, according to ClinicalTrials.gov (https://clinicaltrials.gov, accessed on 28 April 2022).

Retrovirus- and lentivirus-based vectors are commonly used for TCR gene transfer in clinical trials [12]. Both viruses enable stable integration and efficient expression of exogenous TCRs in lymphocytes. However, the mispairing of endo- with exogenous TCRs limits the function of the transduced TCR and generates new antigens, which can cause autoreactivity or GvHD [45]. Nevertheless, there was no evidence of GvHD in the TCR-based adoptive therapy, targeting CHM1 in our treatment trials of EwS, including allogeneic donor lymphocyte infusions [15] or allogeneic transgenic T cells [18]. Several strategies, such as the murinization of TCR constant regions [46,47], codon optimization [48], and insertion of additional cysteine residues [49], have been proposed to prevent mispairing. We used codon optimization and murinization of TCR constant regions. However, these procedures cannot completely eliminate mispairing [50]. The random insertion of viruses into the genome also raises safety concerns, such as insertional mutations and tumorigenesis [13].

### 4.2. Orthotopic Replacement of TCR with Cytotoxic Functionality

To address the potential hazards of viral transduction, endogenous TCR KO with simultaneous non-viral orthotopic TCR replacement has been established. Orthotopic TCR gene replacement drives the translation of the transduced TCR gene sequence, and the activation via the endogenous TCR promoter provides functional results [15,51,52,53].

The comparison of orthotopic TCR gene replacement by CRISPR/Cas9 to random TCR gene insertion by retroviral gene transfer can be performed under different aspects. In clinical application, e.g., in established protocols for production of CAR T cells by retroviral transfer, a multiplicity of infections of the target cell with the retroviral vectors is used, yielding multiple gene copies in the genome of the target cell. In contrast, CRISPR/Cas9-engineered T cells contain only one TCR gene copy. We performed this study, prioritizing the clinical application. However, if a comparison of the biological effects of orthotopic TCR replacement vs. random TCR gene insertion is the aim of study, a MOI is to be chosen under which only a single retroviral insertion of the transgene into the genome of the target cell takes places. This consideration also illustrates that the term transduction efficacy, which is well established in the literature, may be somewhat misleading, since we are measuring yield rather than efficacy when comparing a multiplicity of infections of the target cell with the retroviral vectors to an orthotopic TCR gene replacement into the TRAC locus with a single gene copy by CRISPR/Cas9. Finally, it should be considered that the transduction procedure itself may change the phenotype and CD4/CD8 ratio of T cells.

This present work shows the feasibility of the orthotopic replacement of the endogenous T cell receptor (TCR) with CHM1^319^-TCR targeting EwS by CRISPR/Cas9 and confirms previous publications with different TCRs [15,54]. Our CRISPR/Cas9-engineered T cell products demonstrated a strong specific cytotoxic effect towards HLA-A*02:01+ EwS, comparable to retrovirus-transduced T cells. Clinical application also requires the following considerations: Only patients with relapse or refractory disease after failure of standard therapy are eligible for TCR-based immunotherapy at present. These patients mostly have a short life expectancy and a narrow time window for immunotherapy, which has to be given when patients are, at least, in very good partial remission. In our clinical experience, the production process of therapeutic T cell products often takes too long to provide a clinical benefit within this narrow window. Thus, we were wondering which method yields enough engineered T cells in a short time. Thereby, we had to take into account the low KI efficacy with CRISPR/Cas9 and the high number of cells not surviving electroporation.

Comparing CRISPR/Cas9 with retrovirus-transduced T cells, our work indicates that high retroviral transduction efficacy can prevent endogenous TCR expression on the cell membrane, resembling CRISPR/Cas9-engineered T cells. This may indicate that a high efficiency of TCR transduction by the retrovirus is capable of competing with the endogenous TCR to form the heterocomplex with CD3 required for stable TCR membrane expression. This competition may help to avoid neo-antigen recognition due to TCR chain mispairing. However, high transduction rates may lead to abundant insertion of vector copy numbers (VCN) [55]. According to the reflection paper on clinical risk management, due to insertional mutations from the European Medicines Agency’s Committee on Advanced Therapeutics [56], the risk of gene-modified cell therapies via insertional oncogenesis should be reduced by the restriction of VCN. Additionally, close-to-random transgene integration via viral transduction further limits the clinical application [57]. A low copy number is desired for safety reasons, and it also limits TCR expression, which ultimately compromises the functionality of the T cell product. Taken together, there are limitations by both high- and low-viral transduction rates.

On the other hand, gene editing by CRISPR/Cas9 generates structural defects of the nucleus, chromosomal truncations, micronuclei, and chromosome bridges, which initiate a mutational process and cause human congenital disease, even cancer [58,59]. Rare off-target effects were also identified when using TRAC guide RNA (gRNA) with wild-type Cas9, whereas no off-target effects were detected with the ‘enhanced specificity’ Cas9 variant eSp.Cas9 [53,60]. We performed our experiments by taking advantage of eSp.Cas9, which potentially avoids off-target effects. However, we did not evaluate the genome-wide editing specificity in the present work. For clinical application, we have to ensure the KO of both endogenous TCRs. If the KO of β chain fails, there is a possibility of the mispairing of the transduced α with the endogenous β chain. The ratio of α to β chain in a single cell would be 2:1, since both transgenic chains are expressed from the α locus. Although we do not present TCR sequencing data of CRISPR/Cas9-engineered T cells here, we published detection of transgenic TCRs using specific primers in retrovirally transduced T cells previously [22].

When we compared the transduction efficacy of both procedures, in retrovirally transduced T cells, the amount of transduced TCRs was significantly higher than the endogenous TCR in a single cell, as expected. Thus, the risk of mispairing between exogenous α to endogenous β chains in a single cell would be X:1, with X being >1, depending on the number of transduced gene copies. As expected, retrovirally transduced T cells express more TCR on their surface. However, to strictly compare the biology CRISPR/Cas9 vs. conventional editing, conditions need to be chosen that restrict integration to only one transgene copy. This would allow a strict comparison of orthotopic vs. random TCR placement into the genome. To this end, and to systematically compare CRISPR/Cas9 vs. conventional editing, some of us have recently analyzed expression and functionality of 51 CMV-specific TCRs to show that conventional genetic engineering leads to variable TCR expression and functionality as a result of variable copy numbers with untargeted integration. In contrast, the CRISPR/Cas9-mediated TCR replacement yields homogeneous TCR expression similar to physiological T cells. Product homogeneity after CRISPR/Cas9 TCR gene editing correlated with functionality. Such a well-defined product is obligatory in clinical application [53]. Since efficacies are never certain to be 100%, some of us have also established KI for both chains individually (TCR alpha into TRAC and TCR beta into TRBC), combined with HLA multimer sorting to enrich for the correctly edited population [15].

Finally, we identified a similar amount of CD3 expression on the T cell membrane after CRISPR/Cas9 engineering, compared to non-engineered T cells, while the CD3 expression in retroviral transduced cells was increased, compared to non-engineered T cells. This indicates variable TCR/CD3 complex expression likely depends on the number of TCR gene copies in the genome and the corresponding number of TCR molecules on the T cell membrane.

Several other limitations were also identified in our work, such as comparatively low transduction efficiency of CRISPR/Cas9, ranging from 10–45%, due to the imponderabilities of cultures or fresh vs. thawed status of the cultured lymphocytes. We could minimize cell death after electroporation, especially with TCR KI by directly culturing in Penicillin-Streptomycin (P/S)-free T cell medium after electroporation. We think that higher transduction rates with CRISPR/Cas9-engineered TCR KI are possible. Although cells with a high transduction rate after electroporation do not tolerate the antibiotics (P/S), an optimization of the protocol could involve culturing the cells for 24–48 h in T cell medium without P/S and changing back to the standard culture medium afterwards.

## 5. Conclusions

T cells engineered with CRISPR/Cas9 to address the metastatic driver, CHM1, are feasible for immunotherapy of EwS and may have the advantage of a more prolonged cytotoxic activity, as compared to T cells engineered with retroviral gene transfer, in addition to their homogeneity and functional predictability.

## Figures and Tables

**Figure 1 cancers-14-05485-f001:**
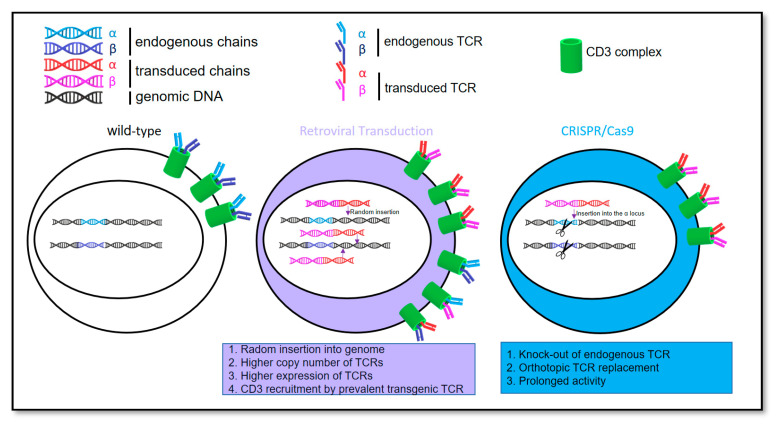
Graphical abstract illustrating the comparison of CRISPR/Cas9-engineered orthotopic TCR replacement or retrovirus transduced random TCR insertion into the T cell genome.

**Figure 2 cancers-14-05485-f002:**
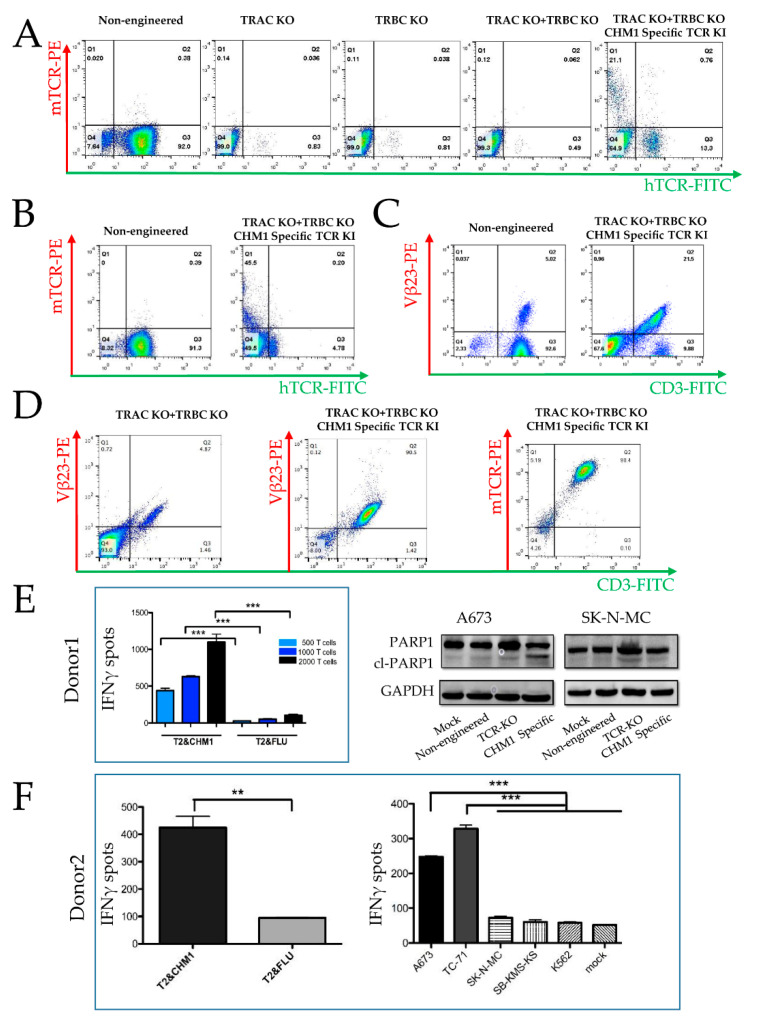
Feasibility of orthotopic TCR replacement by CRISPR/Cas9. (**A**) After knock-out (KO) of either α-chain (TRAC KO) or β-chain (TRBC KO) or of both chains, the endogenous TCR (hTCR) is undetectable by FACS analysis. After KO of both endogenous chains combined with knock-in (KI) of the CHM1^319^-specific TCR, mTCR is detectable (the TRBC constant chain of the CHM1^319^-specific TCR was murinized). Thawed T cells, KI efficiency: 21.3% (thawed T cells). Non-engineered: electroporation of CRISPR/Cas9 but without guide RNA. (**B**) KO of both chains combined with transgenic TCR knock-in: mTCR positive T cells are hTCR negative. Fresh T cells, KI efficiency: 45.5%. (**C**) Pentamer staining confirms transgenic TCR expression as panel B. (**D**) CHM1^319^-TCR expression in the final product of CHM1^319^-TCR insertion after enrichment. (**E**) Donor 1: IFNγ release with dose-dependent manner (500, 1000, and 2000) of transgenic T cells after exposure to T2 cells loaded with either CHM1^319^-peptide or with control-peptide (FLU) for 20 h in donor 1; PARP cleavage (cl-PARP) analyzed by SDS-PAGE after co-culture of A673 (HLA-A*0201+) or SK-N-MC (HLA-A*0201-) with either no T cells (Mock), non-engineered T cells with TCR-KO, or T cells with orthotopic TCR replacement with CHM1^319^ TCR (CHM1-specific), all from donor 1. (**F**) Donor 2: IFNγ release to assess the specific reactivity against several tumor cell lines after co-culture with 1000 CHM1^319^-TCR transgenic T cells in donor 2 (A673 and TC-71: HLA-A*0201+ EwS, SK-N-MC, and SB-KMS-KS: HLA-A*0201- EwS, K562: MHC- NK cell control). IFNγ release transgenic T cells after exposure to T2 cells loaded with either CHM1^319^-peptide or with control-peptide (FLU) serving as positive or negative control. Error bars represent standard deviation of triplicates experiments. ** means *p* < 0.01, *** means *p* < 0.001.

**Figure 3 cancers-14-05485-f003:**
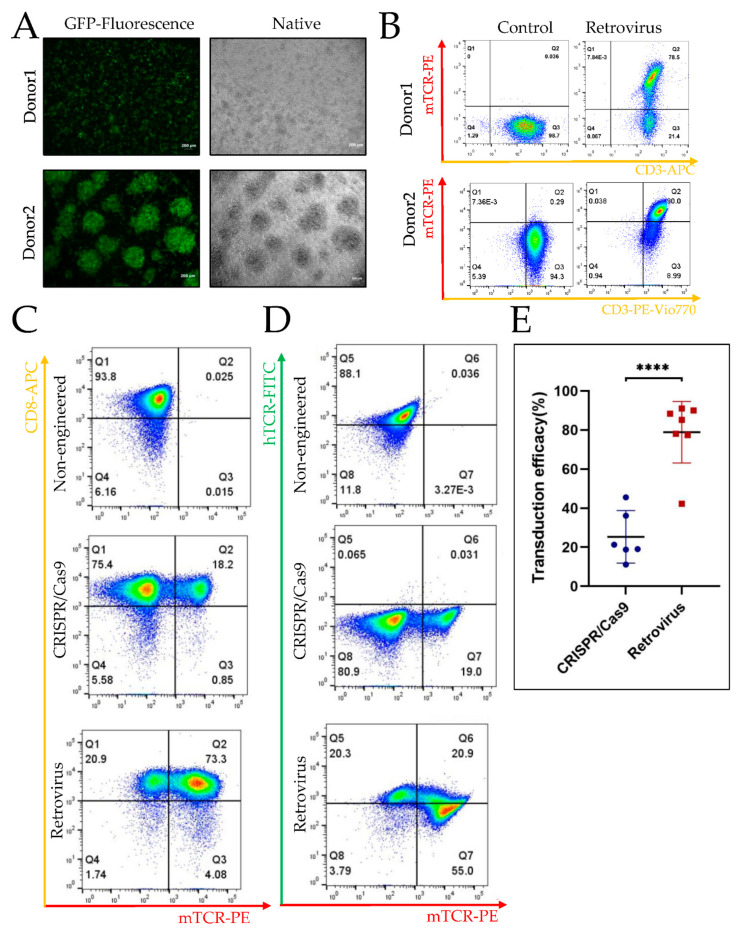
Efficacy of gene editing by CRISPR/Cas9 vs. retroviral transduction, as well as the phenotype of the T cell products. (**A**) A GFP sequence containing pMP71 vector was used to assess transduction efficiency of T cells in general. Transduction is performed twice at day 1 and day 2 of culture. Four days after the first transduction, representative fluorescence microscopy was performed to assess the transduction efficacy of GFP in T cells. Although the colony size is different between the two donors, transduction rates are comparable. Material from donor 1 was thawed; material from donor 2 was fresh. Scale bar: 200 μm. (**B**) The CHM1-TCR sequence containing the pMP71-CHM1-TCR vector was used to transfect T cells. Four days after the first transduction, representative FACS analysis was performed to access the transduction rates. FACS analysis of donor 1 was performed on a Becton Dickinson FACS while analysis of donor 2 was performed on a FACS from Miltenyi. This is the reason for the shift in gating used for donor 1 vs. donor 2. (**C**) T cells were stained with anti-CD8 (CD8-APC) and anti-mTCR (mTCR-PE) after culture to assess the efficacy of transduction. (**D**) T cells were stained with anti-endogenous TCR (hTCR-FITC) and anti-mTCR (mTCR-PE) after culture to assess the efficacy of transduction, the constant domain of the transgenic TCR beta chain being murinized. CRISPR/Cas9 and retrovirus (Retrovirus) transduced T cells from the same donor. (**E**) Transduction efficacy via CRISPR/Cas9 or retrovirus in T cells. **** means *p* < 0.0001.

**Figure 4 cancers-14-05485-f004:**
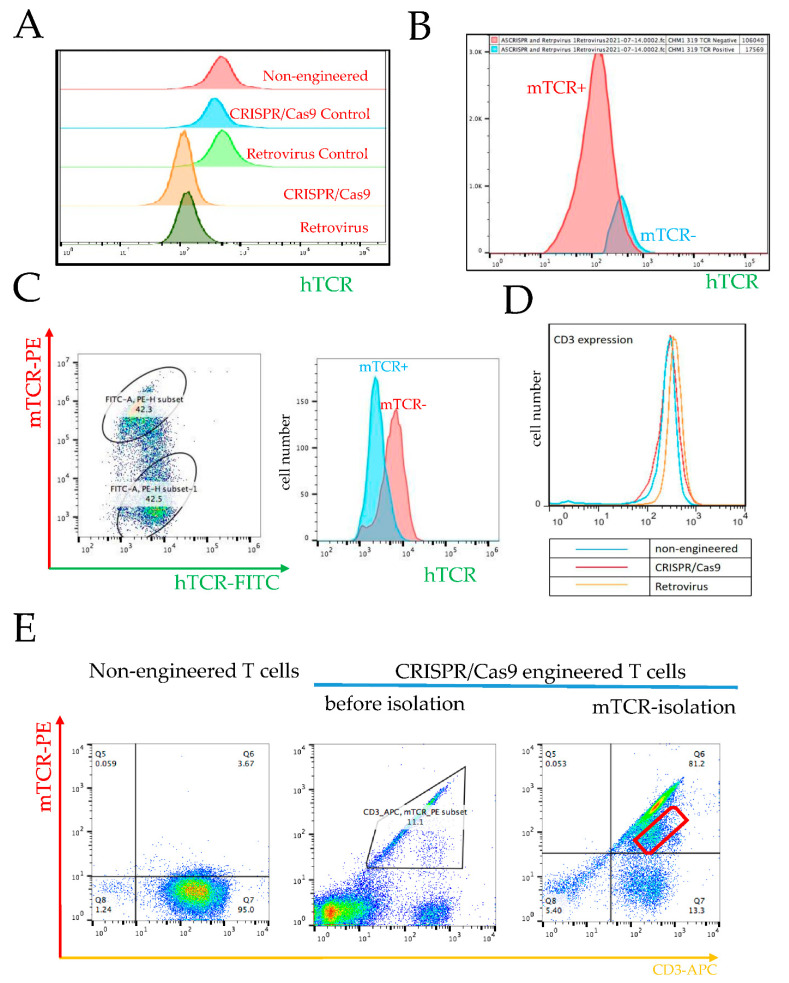
Endogenous TCR expression after gene editing by CRISPR/Cas9 vs. retroviral transduction. (**A**) FACS analysis of expression of endogenous TCR expression (hTCR) from a representative donor: “Non-engineered” designates a control non-engineered T cells; “CRISPR/Cas9 Control” designates a control containing T cells transduced by electroporation of CRISPR/Cas9 but without guide RNA; “Retrovirus Control” designates a control containing T cells exposed to centrifugation only with no retrovirus added; “CRISPR/Cas9” designates T cells with TCR replacement by CRISPR/Cas9, and “Retrovirus” designates T cells with TCR transfer by retrovirus. The “CRISPR/Cas9” and “Retrovirus” groups were analyzed after isolation with anti-mTCR antibody. (**B**) FACS analysis of expression of endogenous TCR expression (hTCR) after retrovirus-mediated TCR transfer (T cells were analyzed before isolation with anti-mTCR antibody). (**C**) FACS determines the expression of endogenous TCR expression (hTCR) on cell membrane after retrovirus-mediated TCR transfer with low efficiency (T cells were analyzed before isolation with mTCR antibody). (**D**) CD3 expression in non-engineered T cells, engineered T cells with orthotopic TCR replacement (CRISPR/Cas9), and engineered T cells retroviral transduction (Retrovirus). (**E**) CRISPR/Cas9-engineered transgenic T cells from thawed donor: left panel, non-engineered T cells; middle panel: CRISPR/Cas9-engineered transgenic T cells before mTCR selection with an anti-mTCR antibody (CRISPR/Cas9); right panel, CRISPR/Cas9-engineered transgenic T cells after mTCR selection with an anti-mTCR antibody. The red cloud in Q6 indicates the transgenic T cell products with a failure KO of endogenous β chain (CRISPR/Cas9).

**Figure 5 cancers-14-05485-f005:**
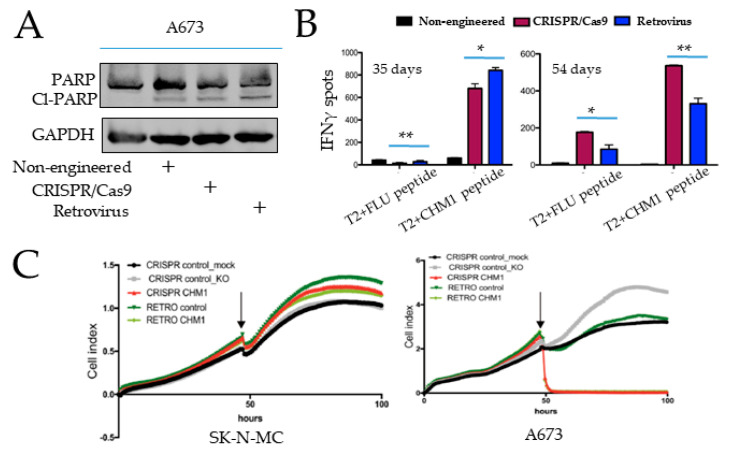
Adoptive transfer of CHM1^319^-TCR transgenic T cells-treated tumor-bearing Rag2-/-γC-/- mice. (**A**) Determination of cl-PARP in A673 (HLA-A*02:01+/CHM1+) cells by SDS-PAGE after co-culture of A673 cells with either no T cells, non-engineered T cells, engineered T cells with orthotopic TCR replacement, or retroviral transduction. (**B**) Evaluation of activation of T cells by IFNγ-ELISpot assay after co-culture with T2 cells plus CHM1 peptide on 35 days (left) and 54 days (right) after T cell isolation from PBMC. A total of 1000 T cells were used in the experiments. (**C**) xCELLigence detachment assays were performed with a different donor to compare the cytotoxic effect of T cells on HLA-A*02:01+/CHM1- SK-N-MC and HLA-A*02:01+/CHM1+ A673 cells. Treatment groups include T cells without CRISPR/Cas9 engineering but otherwise undergoing an identical procedure, including electroporation (CRISPR control_mock), T cells with CRISPR/Cas9-engineered TCR KO only (CRISPR control_KO), T cells with CRISPR/Cas9-engineered TCR KO and TCR replacement (CRISPR CHM1), T cells without retroviral TCR transduction but otherwise identical procedures, including ultracentrifugation (RETRO control) and T cells with retroviral TCR transduction (RETRO CHM1) on day 20. Error bars represent standard deviation of triplicates experiments. * means *p* < 0.05, ** means *p* < 0.01.

## Data Availability

The data of this study can be found in Appendix A.

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
