# Peer review of "T Cells Directed against the Metastatic Driver Chondromodulin-1 in Ewing Sarcoma: Comparative Engineering with CRISPR/Cas9 vs. Retroviral Gene Transfer for Adoptive Transfer"

_cancers, 2022, doi:10.3390/cancers14225485_

Round 1

Reviewer 1 Report

T cells directed against the metastatic driver chondromodulin in Ewing sarcoma: Comparative engineering with CRISPR/Cas9 vs. retroviral gene transfer for adoptive transfer by Busheng Xue et al. addresses a topic of great interest in the field of cancer immunotherapy. The methods compared and the rigor of comparison are highly impressive, the conclusions well supported, and the controls appropriate.

Reviewer 2 Report

The manuscript by Xue et al aims at comparing the effectiveness of two different techniques (CRISPR/Cas9 vs retroviral gene transfer) in the generation of T-cells targeting the metastatic driver chondromodulin-1 in Ewing sarcoma cell lines. The subject is of great interest, however there are several things that need to be addressed which I believe make the manuscript not suitable for publication in its current state.

  Major comments: -it is unclear whether the experiments using PBMCs are from "at least three different donors" (line 128) or "six donors" (line 211) or just two donors, as seen from the experiments shown in figure 2. -I have concerns regarding the compensation parameters used in the FACS analysis on which the entire premise of this work is made. See Figure 2C, 2D, 3D and especially 4E. These should definitely be revised, or at least justified. As far as I can tell, the compensation here is unacceptable -I am unsure about the shift in gating used for donor 1 vs donor 2 in figure 3. What was the rationale behind it? -The arbitrary nature of choosing which are the subgroups in figure 4C without any gating might be hard to understand, I would prefer to see the gating on which percentages can be based on -In figure 2E and 2F two different donors are shown, with different experiments performed (donor 1 PBMCs different concentrations of T cells and PARP cleavage, donor 2 specific reactivity at one unknown T cell concentration and comparison with other cell lines). I think a proper biological replicate with different donors pooled together to show the T cell response to CHM1 would be more scientifically sound. Moreover, why is TC71 not shown in the PARP experiment? I would perform the same experiment with both donor 1 and donor 2 T-cells on A673, SK-N-MC and TC71. Finally, I am not convinced about the PARP1 cleavage: can you provide a quantification of the western blot? If possible, perform another apoptosis experiment such as Annexin V. -are the two donors from figure 2 the same ones from figure 3? If so, would this mean that also in figure 2 donor 1 is thawed while donor 2 is fresh? -Why is the y-axis so different in the IFN spot between donor 1 and donor 2? Is donor 1 more responsive to T2CHM1? -I don't see any significant difference in TCR expression in figure 4D that you claim in line 324. Please perform statistics on the difference in cell number with CD3 expression between non-engineered, CRISPR/Cas9 and retrovirus. -How do you explain that the non-engineered T cells have a similar apoptotic induction compared to the engineered ones, as seen from figure 5A? Please quantify the western blot with three independent experiments. Also, please perform the same experiment with all donors and all EwS cell lines that you have in order to corroborate the data. -Could it be that 54 days is a bit too long and the difference you see is mostly aspecific (figure 5B)?     Minor comments: -at line 270 the opposite of what you claim is mentioned ("As expected, replacement of the endogenous TCR was more efficient in the CRISPR/Cas9...Q5 plus Q6" while showing the opposite in figure 3D). Please change accordingly

Reviewer 3 Report

In their quest for novel therapeutic approaches for Ewing sarcoma (EwS), the highly malignant sarcoma, the authors compared T cells engineered with either CRISPR/Cas9 or retroviral gene transfer. They made three major observations:

1.        Replacing endogenous TCR by CRISPR/Cas9 was feasible and functional

2.        The efficiency of retroviral transduction was higher

3.        CRISPR/Cas9 engineered T cells had prolonged cytotoxic activity

Standard immunology experiments have been applied in the study, which provides a novel insight into immunotherapy based on CRISPR/Cas9 and can interest the large readers of Cancers. The evidence is convincing. I only suggest introducing an additional layer of comparison of the off-target effect. If the author can carry out sequencing of transcriptomes and reveal which approach yields less off-target engineering, it would help with the choice for clinical application.
